# Discrimination of Natural Mature Acacia Honey Based on Multi-Physicochemical Parameters Combined with Chemometric Analysis

**DOI:** 10.3390/molecules24142674

**Published:** 2019-07-23

**Authors:** Tianchen Ma, Haoan Zhao, Caiyun Liu, Min Zhu, Hui Gao, Ni Cheng, Wei Cao

**Affiliations:** 1School of Food Science and Engineering, Northwest University, Xi’an 710069, China; 2School of Chemical Engineering, Northwest University, Xi’an 710069, China; 3Bee Product Research Center of Shaanxi Province, Xi’an 710065, China

**Keywords:** natural mature honey, immature honey, chemometric analysis, multi-physicochemical parameters

## Abstract

Honey maturity is an important factor in evaluating the quality of honey. We established a method for the identification of natural mature acacia honey with eighteen physicochemical parameters combined with chemometric analysis. The analysis of variance showed significant differences between mature and immature acacia honey in physicochemical parameters. The principal component analysis explained 82.64% of the variance among samples, and indicated that total phenolic content, total protein content, and total sugar (glucose, fructose, sucrose) were the major variables. The cluster analysis and orthogonal partial least squares-discriminant analysis demonstrated that samples were grouped in relation to the maturity coinciding with the results of the principal component analysis. Meanwhile, the 35 test samples were classified with 100% accuracy with the method of multi-physicochemical parameters combined with chemometric analysis. All the results presented above proved the possibility of identifying mature acacia honey and immature acacia honey according to the chemometric analysis based on the multi-physicochemical parameters.

## 1. Introduction

Acacia honey is the natural sweet substance produced by honeybees, which collect nectar from the flowers of *Robinia pesudoacacia* (Figure 1), transform and combine it with specific substances of their own, store it, and leave it in the honeycomb to ripen and mature [1]. In the process of maturation, honey properties and chemical compositions are also changed as a result of biotransformation, biodegradation, and bioaccumulation, which have a great influence on the quality and authenticity of natural honey [2,3,4,5]. Foraging bees collect nectar through their proboscis, and place it in the proventriculus (honey stomach) [4,6,7,8,9]. Meanwhile, proteins and salivary enzymes from the hypopharyngeal glands of bees begin breaking down sugars from the nectar [7,10,11,12,13]. Hydrogen peroxide and gluconic acid, formed by the degradation of glucose oxidase, are partly able to suppress bacterial growth and are responsible for increasing the acidity of honey [14,15]. As nectar collection is completed, the hive bees continually digest nectar and hydrolyze sucrose into glucose and fructose by using bee digestive enzymes [2,7,8,16,17,18,19]. Glucose and fructose indicate about 75% of the sugars found in honey, which play important roles in honey quality control and authenticity. The ratio between fructose and glucose, as well as their concentration, are commonly used for predicting honey crystallization and are a beneficial index for the classification of monofloral honeys [20,21]. Another major transformation that occurs during the maturity process is moisture evaporation. The hive bees store digested nectar in the honeycomb cells and transfer it from one cell to another. Hive bees flutter their wings continuously to circulate air and to evaporate moisture from the honey to about 18% and eventually cover the cells with wax to seal them [2,7,22]. Reduction of moisture content below 18% is deemed to be a secure level for retarding yeast activity, decreasing the rate of fermentation and avoiding the appearance of undesirable flavor [23,24,25,26,27]. According to all the above, honey with less than 18% moisture, a sugar concentration above the saturation point, and sealed honeycomb cells may be considered as natural mature honey. 

Because of the influence of components change during the honey maturation process, natural mature honey has its own characteristics, which differ from immature honey. Biologically speaking, immature honey lacks many of the positive properties of natural mature honey. Natural mature honey is a highly complex product with around 200 different substances, which cannot be artificially emulated [28,29]. Physicochemical parameters can provide useful and complete information for the composition and properties of honey and are effective to assess the honey quality and authenticity [30,31,32,33]. Chemometric methods can lessen the complexity of large data sets and offer better explication and construction of data sets, as well as identify the natural clustering pattern and group variables based on similarities between samples [34,35,36]. Combining the great deal of data acquired from physicochemical parameters with chemometrics may be an excellent measure for discrimination of natural mature honey.

The purpose of our research is to identify natural mature honey by establishing the method of multi-physicochemical parameters combined with chemometric analysis. Our study stands out as the first report comparing mature honey with immature honey using principal component analysis (PCA), cluster analysis (CA), and orthogonal partial least squares-discriminant analysis (OPLS-DA). Mature honey was differentiated by evaluating the similarities and discriminant features of honey samples qualitatively, a method that is expected to be applicable to quality control and authenticity identification of natural mature honey.

## 2. Materials and Methods

### 2.1. Honey Sample

A total of 85 acacia (*Robinia pseudoacacia*) honey samples were collected from several geographical areas of Shaanxi, China (Table 1) and were kept at 4 °C prior to analysis. The botanical origin of the samples was confirmed by the method of Lutier and Vassiere [37]. All honey samples were collected from a specific mature honey demonstration basement (Figure 1). The natural mature acacia honey (A1–A29) was collected from honey that was capped in the hive and brewed by bees for 7 to 10 days. Immature acacia honey (B30–B85) was collected by hive bees that brewed honey for one to three days. In addition, all honey samples were extracted using a honey extractor and filtered to remove beeswax and other debris.

The honey samples numbered A16–A29, B37–B45, B46–B53, B56–B58, B61–B63, B66–B68, B71–B73, B76–B78, and B81–B83 were randomly selected for the calibration set, and the remaining 35 samples were used as test samples to verify the accuracy of this method.

### 2.2. Pollen Analysis

The botanical origin of the samples was determined using the method of Lutier and Vassiere [37]. For floral identification, honey samples (5 g) were thoroughly mixed with distilled water (5 mL), and centrifuged at 3000 rpm for 10 min, to separate the pollens. Samples of separated pollen grains were spread with the help of a brush on a slide containing a drop of lactophenol. The slides were examined microscopically at 45× magnification, using a bright-field microscope (Olympus, Tokyo, Japan). According to the different volumes, contours, grooves, holes, and other characteristics of pollen morphology, as well as pictures of different varieties, pollen varieties were identified. A total of 40 horizons and a certain number of pollen grains were observed (the total number of pollen should be more than 100 grains).
$$Pollen\;Content\;\left(\%\right)\;=\frac{a\;certain\;pollen\;number\;of\;40\;horizons}{a\;total\;pollen\;number\;of\;40\;horizons}\;\times \;100\%$$

### 2.3. Physicochemical Properties

The methods used for the quantitative analysis of physicochemical properties were determined mainly according to the Association of Official Analytical Chemists (AOAC) [38]. The details of the methods used in this study are summarized in Appendix A.

### 2.4. HPLC Conditions

The contents of fructose, glucose, and sucrose were determined by high-performance liquid chromatography (HPLC) and a refractive index detector (Shodex R1-201H, Shanghai, China). The column was a Waters carbohydrate high performance (4.6 × 250 mm, 4 μm; Waters). Honey samples (5 g) were thoroughly mixed with ultrapure water (60 mL), and the total volume of the mixture was adjusted to 100 mL with acetonitrile. The mobile phase was 78% acetonitrile and 22% ultrapure water (*v*/*v*), using an isocratic method. The solutions were filtered through a 0.45 μm membrane filter prior to use. The column was operated at 35 °C, the detector pool temperature was 35 °C, the flow-rate was 1.0 mL min^–1^, and the injection volume was 15 μL. 

### 2.5. Data Analysis

The multivariate statistical analysis was analyzed by SIMCA (Version 14.1, Umetrics, Umeå, Sweden). The main chemometrics methods used were principal component analysis (PCA), cluster analysis (CA), and orthogonal partial least squares-discriminant analysis (OPLS-DA). PCA simplifies multiple indexes into a small number of comprehensive indexes and uses as many variables as possible to reflect the information of the original variables [39]. CA is the aggregation of samples according to the similarity degree of quality characteristics, and the most similar priority polymerization [40,41]. OPLS-DA is a regression modeling method from multiple dependent variables to multiple independent variables [42]. CA and PCA were used to analyze the comprehensive change of honey, with mean-centered, UV scaled, and log-transformed data before building the PCA model. OPLS-DA was carried out to discriminate features with mean-centered, Pareto scaled and log-transformed data, and the validation of the model was tested using seven-fold internal cross-validation and permutation tests for 200 times. Significant differences were determined using Mann–Whitney U tests, and *p* < 0.05 was considered to be statistically significant. In order to avoid the influence of moisture on other variables, we preprocessed the variables and adopted the dry-weight value of each variable, that is, normalized the moisture content of the variables, and used the ratio of the actual variable value to the content of moisture as the dry weight value of the corresponding variables.

## 3. Results and Discussion

### 3.1. Pollen Analysis

Table 1 shows the floral origin of acacia honeys determined by microscopy pollen analysis. The data indicate that all the honey samples were monofloral. *Robinia pseudoacacia* pollen was detected in all samples more than 80.00%.

### 3.2. Physicochemical Parameters Analysis

Table 2 shows the mean values of the physicochemical parameters of natural mature acacia honey (NMH) and immature acacia honey (IMH) from different regions of the Shaanxi Province. Comparing the physicochemical parameters of NMH and IMH, we found significant differences between NMH and IMH in the mean values of moisture, sugars, protein content, total phenolic, and proline (Figure 2).

Moisture is an important standard for evaluating honey quality, as it can determine the shelf life of honey and its ability to resist fermentation deterioration [43]. The European and Codex standards established a limit of 20% in the case of honey, which can be kept for long periods of time without becoming spoiled [2,9,27]. The moisture of NMH ranged from 15.63% to 16.61%, and IMH was between 21.14% and 26.61%, exceeding the limit, making it difficult to store, producing acids and alcohols more easily, and seriously affecting the quality of honey [5,20].

Honey is a saturated solution of sugars, which accounts for about 70–75% of soluble sugar. Fructose and glucose account for the largest proportion of honey composition, but a small quantity of sucrose was also discovered [25,30]. They are the building blocks of more complex sugars such as disaccharides and maltose [33]. The content was within the limits of European and Codex standards of 65% minimum for glucose and fructose, where sucrose content should be not more than 5%. The sugar concentration (fructose and glucose) of NMH was approximately from 74.01% to 84.06%, which confirmed that samples were genuine honeys [30,32,44]. 

The total protein content of NMH was between 510.49 mg/kg and 622.29 mg/kg and that of IMH was lower (369.79–538.35 mg/kg). It is universally known that honey contains a trace amount of protein, usually formed by bees, which constantly swallow honey and flutter their wings [31]. The variability in the protein content of different maturity of types honey may be related to its brewing time and brewing degree [45]. 

During the ripening process, enzymes are proteins that help to speed up many chemical reactions in living organisms, and to convert certain substances into different products [46], especially diastase, invertase, and glucose oxidase. Owing to the high enzyme activity, NMH showed a richer nutritional value than IMH.

The electrical conductivity (EC) is based on the acid contents and ash of honey [27,31], and the free acidity is the content of free acids, which is determined by the equivalence point titration [47]. Furthermore, the pH is correlative with the stability and the shelf life of honey. The pH values of honey usually range from 3.5 to 5.5 [30,33]. The results of statistical analysis were not significantly different in electrical conductivity, pH, and free acidity. 5-hydroxymethylfurfural (HMF) is considered to be a useful indicator for heat treatment and long-term storage of honey [36]. An excessive amount of HMF has been considered evidence of overheating, characterized by a darkening of color and a loss of freshness of honey [23]. In this study, HMF was found to be below the detection limit, indicating that all samples were fresh and not overheating.

### 3.3. Principal Component Analysis (PCA)

PCA is a statistical method that simplifies several related indicators into a few comprehensive indicators [48]. PCA was combined with the physicochemical parameters in Table 2, in order to avoid the influence of moisture on other physicochemical parameters. The moisture was eliminated when the model was established, and the data of other parameters were normalized and used on a dry-weight basis. The principal component analysis results of the 17 indicators for 50 samples are shown in Table 3. There were three principal components, which are in agreement with the results in Table 3.

We standardized data to ensure that all the elements had an equal influence over the results. The eigenvalues and the percentage variance, explained by principal components, are shown in Table 3. Three components, eigenvalues > 1, were extracted and used to examine the dataset. The first three components accounted for 82.64% of the total variance. Principal component 1 (PC1) expressed 60.05% of the variance, and the next principal components explained 13.79% and 8.81% of the variance, respectively. The loadings of each compound on the principal component analysis explicitly showed that the grouping of the different maturity honey was mainly influenced by certain compounds. PC1 and PC2 of all the samples explained 73.83% of the total variance at length. PC1 was directly relevant to L*, total phenolic content, and proline, and the dominant variables were mainly affected by protein, total phenol, and sucrose in PC2 (Figure 3B: loading). Moreover, honey samples were properly classified, NMH was classified into one category, and IMH was separated into another category (Figure 3A: scores of sample). In summary, mature honey samples are significantly different from immature honey, which is in agreement with the result of the physicochemical parameters analysis.

### 3.4. Cluster Analysis

Cluster analysis is a kind of statistical method used to classify the characteristics of multi-indicators and multi-objects [34]. It gradually aggregates according to the similarity of sample quality characteristics. The greatest degree of similarity is aggregated, and multiple varieties are integrated according to the comprehensive nature of the categories [41]. In this study, based on the physicochemical parameters, 50 samples were subjected to the Ward method. The cluster analysis of the pedigree chart is shown in Figure 4. Acacia honey samples were separated into two categories. The first category contained NMH and the second group, namely the remaining samples, contained IMH. The distance between these two categories was more than 100.

According to the CA and the PCA, all honey samples were divided into two types, which were basically consistent with the analysis of the parameters mentioned above: color, electrical conductivity, pH, free acid, lactone, total acid, glucose, fructose, sucrose, total sugar content, total phenolic content, amylase activity, HMF, total protein content, glucose oxidase, and other basic physicochemical parameters.

### 3.5. Orthogonal Partial Least Squares-Discriminant Analysis

Orthogonal partial least squares-discriminant analysis (OPLS-DA) was developed to distinguish between NMH and IMH, and score plots of these models were applied for separating samples. The cross-validation method was used to verify the model. A total of three principal components were selected. The fitted model’s index R^2^X (cum) was 0.928, indicating that the four principal components explained 92.80% of the X variables, and the index of fitting the dependent variable R^2^Y (cum) was 0.978, indicating that the four principal components interpreted 97.80% of the Y variable. The model prediction index Q^2^ (cum) was 0.971, explaining that the model had a predictive power of 97.10% for mature honey and immature honey and that this model was stable and reliable. Permutation was conducted 200 times in order to further test the predictability of the OPLS-DA model. To show the predictability, all R^2^ and Q^2^ (Figure 5B) were > 0 and <−0.5, respectively.

An S-plot was adopted to visualize the influence of sensitive indices on the mature and immature honey (Figure 5A). Discriminant markers are located in the upper right and lower left corners of the S-plot, with higher absolute p [1] and p (corr) values to select the potential markers [49]. In this model, the sensitive markers were proline, total protein content, EC, L*, a*, b*, and total phenolic content. The variable importance to projection (VIP) values of these indicators are more important than 1. Figure 5 demonstrates a direct separation between NMH and IMH.

### 3.6. Test Samples Analysis

Thirty-five samples were tested. The results of the PCA score plots (Appendix A) show that NMH samples were classified into one category and IMH samples into another. We took distance as a dependent variable and the test samples as independent variables for cluster analysis. Appendix A shows that 35 test samples were properly classified. The accuracy of both the method and cross-verification is 100%.

The score plots of the OPLS-DA models were developed for separating test samples. The R^2^X of test honey supervised models was 0.901, R^2^Y was 0.985, and Q^2^ calculated from seven-fold cross-validation was 0.978. All R^2^ and Q^2^ were > 0 and <−0.5, respectively, showing a predictability. The classification results are shown in Appendix A. Thirty-five test samples were correctly classified on the basis of their maturity. The overall correct classification rate of the original and cross-validation methods is 100%.

Both the 50 honey samples and the 35 test samples achieved appropriate classification results according to the physicochemical parameters combined with chemometric methods. Therefore, physicochemical parameters combined with chemometric methods could be used in the classification of NMH and IMH.

## 4. Conclusions

In conclusion, the current results clearly show that there are significant differences in physicochemical parameters between natural mature acacia honey and immature acacia honey. Principal component analysis showed that total phenolic content, total protein content, and total sugar (glucose, fructose, sucrose) were the main parameters affected seriously by honey maturity. The cluster analysis and orthogonal partial least squares-discriminant analysis showed that samples were grouped in relation to the maturity and the overall correct classification rate reached 100%. The approach of multi-physicochemical parameters combined with chemometrics is effective in discriminating natural acacia mature honey. However, to confirm the applicability of the other monofloral honeys, this approach needs to be validated. We look forward to the development of this as a promising method for authenticity identification of natural mature honey in any quality control of businesses and government agencies. 

## Figures and Tables

**Figure 1 molecules-24-02674-f001:**
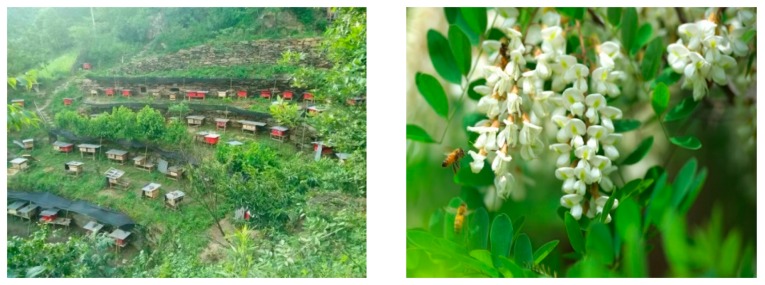
Photograph of mature honey demonstration basement and acacia tree flower.

**Figure 2 molecules-24-02674-f002:**
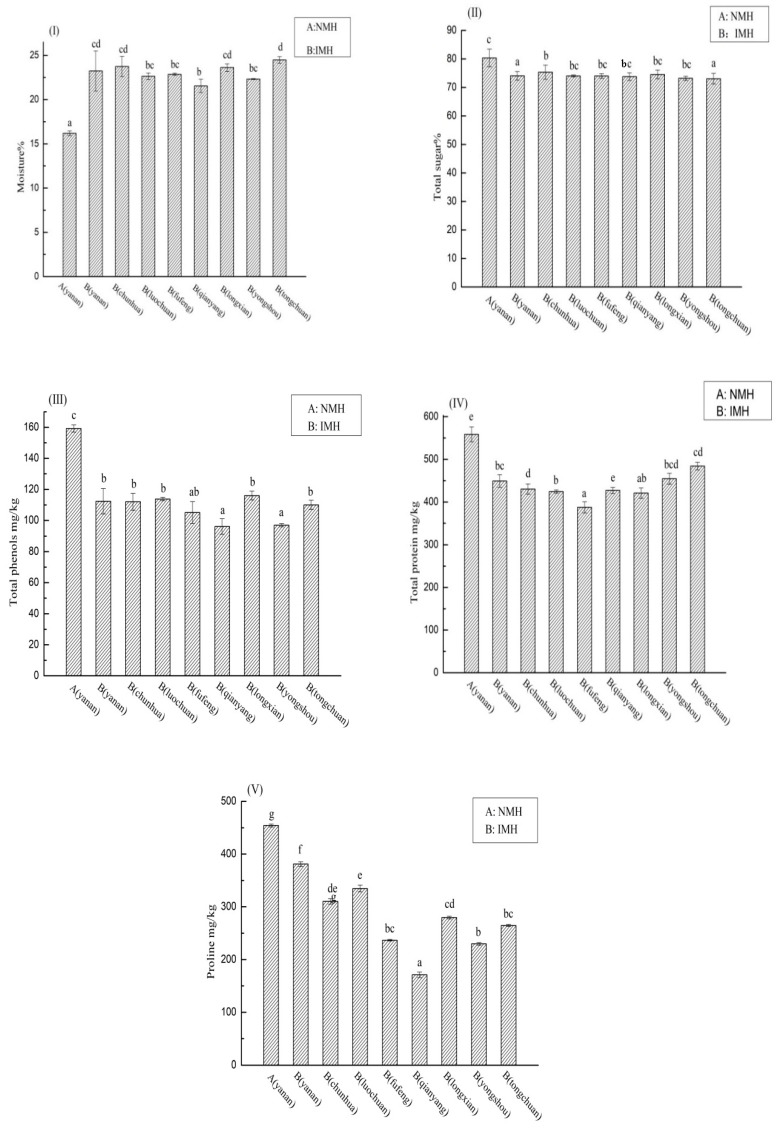
Significant physicochemical parameters of natural mature acacia honey (NMH) and immature acacia honey (IMH). Different lower case letters correspond to significant differences at *p* < 0.05. (I) represents moisture; (II) represents total sugar content; (III) represents total phenols content; (IV) represents total protein content; (V) represents proline content.

**Figure 3 molecules-24-02674-f003:**
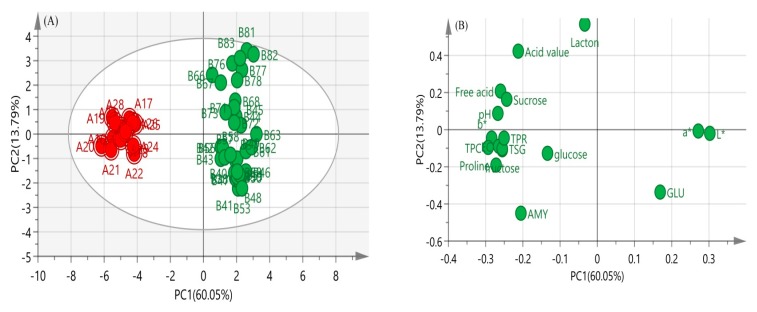
The principal component analysis (PCA) score plots (**A**) and PCA loading plots (**B**) of acacia honey samples. GLU: glucose oxidase; TSG: total sugar; AMY: amylase activity; EC: conductivity; TPC: total phenolic content; TPR: total protein content (for interpretation of the references to color in this figure legend, the reader is referred to the Web version of this article).

**Figure 4 molecules-24-02674-f004:**
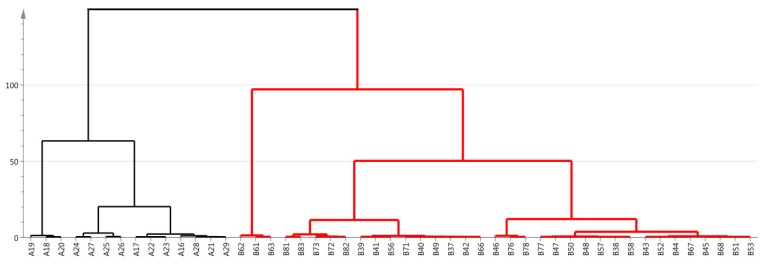
Results of hierarchical cluster analysis of samples.

**Figure 5 molecules-24-02674-f005:**
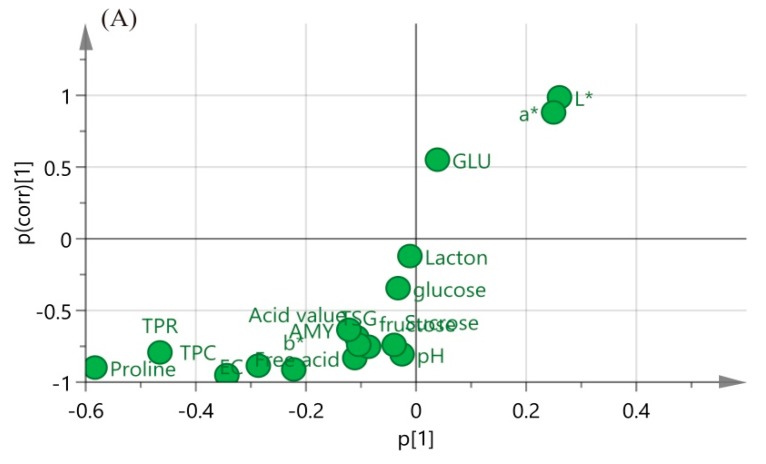
The orthogonal partial least squares-discriminant analysis (OPLS-DA) S-plots (**A**), validation plot (**B**), and score plots (**C**) of honey samples. P [1] is the loading vector of covariance in the first principal component. P (corr) [1] is loading vector of correlation in the first principal component. Variables with |*p*| ≥ 0.05 and |*p* (corr)| ≥ 0.5 are considered statistically significant. R^2^ is the fitted model’s index; Q^2^ is the model prediction index. GLU: glucose oxidase; TSG: total sugar; AMY: amylase activity; EC: conductivity; TPC: total phenolic content; TPR: total protein content (for interpretation of the references to color in this figure legend, the reader is referred to the Web version of this article).

**Table 1 molecules-24-02674-t001:** Characterization of the analyzed acacia honey samples.

Samples	Type of Honey	Botanical Source	Production Region	Predominant Pollen (%)
A1–A29	Monofloral	*Robinia pseudoacacia*	Yan’an, Shaanxi	88.38 ± 2.56
B30–B45	Monofloral	*Robinia pseudoacacia*	Yan’an, Shaanxi	87.21 ± 1.78
B46–B55	Monofloral	*Robinia pseudoacacia*	Chunhua, Shaanxi	86.31 ± 3.01
B56–B60	Monofloral	*Robinia pseudoacacia*	Luochuan, Shaanxi	82.61 ± 1.89
B61–B65	Monofloral	*Robinia pseudoacacia*	Fufeng, Shaanxi	85.77 ± 3.21
B66–B70	Monofloral	*Robinia pseudoacacia*	Qianyang, Shaanxi	83.60 ± 2.18
B71–B75	Monofloral	*Robinia pseudoacacia*	Longxian, Shaanxi	84.34 ± 1.63
B76–B80	Monofloral	*Robinia pseudoacacia*	Yongshou, Shaanxi	88.63 ± 3.84
B81–B85	Monofloral	*Robinia pseudoacacia*	Tongchuan, Shaanxi	86.67 ± 1.67

**Table 2 molecules-24-02674-t002:** Physicochemical property of samples (dry-weight basis).

**Samples**	**L***	**a***	**b***	**Conductivity (µS/cm)**	**pH**	**Free Acid (meq/kg Dry Matter)**	**Lacton (meq/kg Dry Matter)**	**Acid Value (meq/kg Dry Matter)**	**HMF (mg/kg Dry Matter)**
A1–A29	47.60 ± 2.82 ^a^	101.99 ± 4.30 ^a^	23.49 ± 6.72	136.25 ± 3.38 ^g^	3.17 ± 0.06 ^c^	27.17 ± 1.72 ^c^	5.98 ± 0.99 ^a,b^	33.15 ± 1.58 ^d^	N.D
B30–B45	70.99 ± 1.53 ^b,c^	124.24 ± 2.29 ^b^	8.79 ± 2.73 ^a^	108.66 ± 5.58 ^c,d^	2.88 ± 0.08 ^a,b^	21.21 ± 1.09 ^a,b^	4.94 ± 1.71 ^a^	26.15 ± 2.38 ^a,b^	N.D
B46–B55	70.71 ± 0.71 ^b,c^	125.64 ± 1.71 ^b^	3.62 ± 0.08 ^a^	103.70 ± 1.58 ^c^	2.89 ± 0.07 ^a,b^	20.27 ± 0.74 ^a^	3.78 ± 0.57 ^a^	24.05 ± 0.94 ^a^	N.D
B56–B60	72.95 ± 0.86 ^d^	123.25 ± 2.41 ^b^	5.92 ± 0.35 ^a^	113.88 ± 0.67 ^d,e^	2.94 ± 0.05 ^a,b^	22.77 ± 0.25 ^b^	4.76 ± 0.49 ^a^	27.54 ± 0.25 ^b^	N.D
B61–B65	72.25 ± 1.07 ^c^	129.64 ± 1.88 ^b^	4.01 ± 0.29 ^a^	110.71 ± 3.93 ^c,d^	2.89 ± 0.04 ^a,b^	23.04 ± 0.86 ^b^	4.86 ± 0.49 ^a^	27.91 ± 1.31 ^b,c^	N.D
B66–B70	71.45 ± 0.55 ^b,c^	125.50 ± 1.65 ^b^	3.75 ± 0.13 ^a^	84.61 ± 2.25 ^a^	3.09 ± 0.03 ^c^	23.83 ± 0.49 ^b^	8.79 ± 0.55 ^c^	31.62 ± 1.04 ^d^	N.D
B71–B75	69.48 ± 0.44 ^b,c^	116.85 ± 0.58 ^b^	6.29 ± 0.21 ^a^	117.68 ± 4.80 ^f^	2.83 ± 0.02 ^a^	22.94 ± 0.83 ^b^	7.98 ± 0.35 ^b,c^	30.92 ± 0.84 ^c,d^	N.D
B76–B80	71.65 ± 0.16 ^b,c^	130.71 ± 1.11 ^b^	2.56 ± 0.17 ^a^	93.43 ± 2.15 ^b^	2.99 ± 0.02 ^b^	23.48 ± 0.43 ^b^	7.35 ± 0.63 ^b,c^	30.83 ± 0.98 ^c,d^	N.D
B81–B85	68.34 ± 0.14 ^b^	120.92 ± 1.44 ^b^	3.73 ± 0.21 ^a^	105.69 ± 1.09 ^c^	2.86 ± 0.02 ^a,b^	23.25 ± 0.59 ^b^	7.75 ± 0.19 ^b,c^	31.00 ± 0.44 ^c,d^	N.D
**Samples**	**Glucose (g/100 g Dry Matter)**	**Fructose (g/100 g Dry Matter)**	**Sucrose (g/100 g Dry Matter)**	**Total Sugar (g/100 g Dry Matter)**	**Total Phenolic (mg/kg Dry Matter)**	**Total Protein (mg/kg Dry Matter)**	**Amylase Activity (^o^ Gothe)**	**Proline (mg/kg Dry Matter)**	**Glucose Oxidase (U/g Dry Matter)**
A1–A29	26.04 ± 1.22 ^b,c^	37.39 ± 1.12 ^c^	1.97 ± 0.26 ^d^	65.40 ± 1.71 ^c^	126.59 ± 7.85 ^c^	454.09 ± 11.48 ^f^	39.15 ± 2.44 ^f^	343.35 ± 11.42 ^f^	1.26 ± 0.21 ^a^
B30–B45	25.20 ± 0.36 ^b^	34.07 ± 0.50 ^b^	1.17 ± 0.17 ^a,b^	60.44 ± 0.58 ^b^	89.06 ± 1.21 ^b^	361.14 ± 10.13 ^c,d^	33.86 ± 1.65 ^d,e^	238.21 ± 11.65 ^d^	2.36 ± 0.18 ^b^
B46–B55	25.18 ± 0.24 ^b^	33.93 ± 0.73 ^b^	1.15 ± 0.11 ^a,b^	60.26 ± 0.66 ^b^	87.81 ± 2.46 ^b^	388.65 ± 2.28 ^d,e^	35.50 ± 1.23 ^e^	232.51 ± 1.54 ^d^	2.29 ± 0.30 ^b^
B56–B60	25.61 ± 0.45 ^b,c^	34.07 ± 0.43 ^b^	1.43 ± 0.03 ^b,c^	61.11 ± 0.10 ^b^	88.71 ± 1.12 ^b^	329.02 ± 3.11 ^a,b c^	33.34 ± 1.03 ^d,e^	266.43 ± 13.60 ^e^	2.53 ± 0.16 ^b^
B61–B65	24.67 ± 0.53 ^b^	34.11 ± 0.87 ^b^	1.14 ± 0.15 ^a,b^	59.93 ± 0.59 ^b^	79.35 ± 6.51 ^a,b^	300.09 ± 11.81 ^a^	29.62 ± 1.60 ^b,c^	183.34 ± 0.41 ^b^	3.25 ± 0.61 ^c^
B66–B70	27.28 ± 1.14 ^c^	35.61 ± 0.53 ^b^	1.41 ± 0.15 ^b,c^	64.29 ± 1.15 ^c^	74.78 ± 4.65 ^a^	416.62 ± 8.67 ^e^	27.61 ± 1.38 ^b^	133.24 ± 3.41 ^a^	1.96 ± 0.58 ^b^
B71–B75	24.89 ± 0.59 ^b^	34.31 ± 0.83 ^b^	0.89 ± 0.11 ^a^	60.09 ± 1.48 ^b^	89.39 ± 2.16 ^c^	319.56 ± 12.93 ^a,b^	31.41 ± 0.68 ^c,d^	213.51 ± 2.637 ^c^	2.37 ± 0.86 ^b^
B76–B80	24.88 ± 0.31 ^b^	34.32 ± 0.13 ^b^	1.68 ± 0.46 ^c,d^	60.88 ± 0.68 ^b^	74.97 ± 0.89 ^a^	350.63 ± 12.13 ^b,c d^	21.40 ± 0.59 ^a^	179.95 ± 0.32 ^b^	1.22 ± 0.21 ^a^
B81–B85	22.46 ± 1.86 ^a^	30.52 ± 1.79 ^a^	1.54 ± 0.07 ^b,c^	54.52 ± 1.67 ^a^	82.92 ± 2.77 ^a,b^	366.10 ± 8.53 ^c,d^	21.82 ± 1.03 ^a^	199.24 ± 1.97 ^b,c^	0.66 ± 0.15 ^a^

A1–A29: mature honey; B30–B85: immature honey; N.D: not detected. Results presented in the table are expressed as the mean values ± standard deviation (SD). Different lower case letters correspond to significant differences at *p* < 0.05.

**Table 3 molecules-24-02674-t003:** Variance contribution rate and composition load matrix. PC, principal component.

Quality Index	PC1	PC2	PC3
L*	−0.963	0.029	0.153
a*	−0.870	0.014	0.175
b*	0.905	0.069	−0.134
Conductivity (μS/cm)	0.847	0.142	−0.333
pH	0.852	−0.142	0.218
Free acid (meq/kg)	0.829	−0.314	−0.034
Lacton (meq/kg)	0.106	−0.875	0.240
Acid value (meq/kg)	0.676	−0.651	0.088
Glucose %	0.425	0.195	0.785
Fructose %	0.817	0.164	0.338
Sucrose %	0.777	−0.252	−0.113
Total sugar %	0.814	0.154	0.533
Total phenolic content (mg/kg)	0.936	0.146	−0.160
Total Protein (mg/kg)	0.798	0.073	0.045
Amylase activity (^o^Gothe)	0.654	0.692	0.008
Proline (mg/kg)	0.872	0.294	−0.301
Glucose oxidase (U/g)	−0.538	0.516	0.234
Eigenvalues	10.208	2.343	1.497
Contribution rate %	60.045	13.785	8.806
Cumulative contribution%	60.045	73.830	82.636

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
