# Peer review of "Discrimination of Natural Mature Acacia Honey Based on Multi-Physicochemical Parameters Combined with Chemometric Analysis"

_molecules, 2019, doi:10.3390/molecules24142674_

Round 1
Reviewer 1 Report
Dear Authors,
Please address the following issues before publication.
Major issues.
Significance of differences in parameters between mature and immature samples should be checked by pooling all sampling sites together (one group immature vs one group mature). That can be meaningfully tested with Mann-Whitney test. Provide p-values.
Minor issues.
L37: proventriculus?
L84-86: please verify that the sample numbering is correct
Table 1: there are two types of fonts, please correct
L189: did you mean “acidity correlates with the microbiological stability” (?)
L204: you plotted/used three, but there should be many PCs
Table 2 header: lactone
Fig.2. lowercase letters showing significance seems to be mixed in subplot II. Why is IMH c, just like the sixth sample?
L243: dispersal peace ? please revise.
Include the used statistical test in Table headers and Figure captions, where relevant.
Best regards.
Author Response
We are truly grateful to your critical comments and thoughtful suggestions.
Reviewer:
Significance of differences in parameters between mature and immature samples should be checked by pooling all sampling sites together (one group immature vs one group mature). That can be meaningfully tested with Mann-Whitney test. Provide p-values.
Minor issues.
Question 1: L37: proventriculus?
Response: Thank you for your comment. The mistake has been corrected in line 37.
Question 2: L84-86: please verify that the sample numbering is correct.
Response: Thank you for your comment. We have carefully checked the sample number, and we have a total of 29 mature honey samples (A1-A29) and 56 immature honey samples (B30-B85).
Question 3: Table 1: there are two types of fonts, please correct
Response: Thank you for your comment. The mistake has been corrected in Table 1.
Question 4: L189: did you mean “acidity correlates with the microbiological stability” (?)
Response: We appreciate your comment. The natural honey contains a variety of organic acids and inorganic acid, and the pH values vary slightly with the type of honey, but usually range from 3.5 to 5.5. There is a limit to the tolerance of microorganisms to acidity and alkalinity, beyond which metabolism will slow down or even stop. Generally, the optimum pH value for microbiological growth is about 7, and the suitable growth pH ranges for Staphylococcus, Streptococcus, Brucella, Escherichia, Pasteurella, and Bacillus are 7-7.5, 7.4-7.6, 6.6-7.4, 7.2-7.4, 7.2-7.4, and 7.2-7.6. The acid environment of honey is obviously not suitable for the growth of microorganism, and can inhibit or eliminate the growth of microorganism.
Pita-Calvo C, Guerra-Rodriguez M. E, Vazquez M (2017) A review of the analytical methods used in the quality control of honey. Journal Agriculture Food Chemistry 65(4) 690-703, https://doi.org/ 10.1021/acs.jafc.6b04776
Silva P. M. D, Gauche C, Gonzaga L. V (2016) Honey chemical composition, stability and authenticity. Food Chemistry 196 309-323, https://doi.org/ 10.1016/j.foodchem.2015.09.051
Popek S, Halagarda M, Kursa K (2017) A new model to identify botanical origin of polish honeys based on the physicochemical parameters and chemometric analysis. LWT-Food Science and Technology 77 482-487, https://doi.org/ 10.1016/j.lwt.2016.12.003
Question 5: L204: you plotted/used three, but there should be many PCs
Response: Thank you for your comment. PCA is a multivariate statistical analysis method to extract information from the original data and reduce the dimensionality while simplifying the data. In data mining, eigenvalues are directly used to describe the amount of information contained in the direction of the corresponding eigenvector, and the sum of all eigenvalues divided by a certain eigenvalue is: the variance contribution rate of the eigenvector (the variance contribution rate represents the proportion of the information contained in the dimension). Usually, the data changed by eigenvector is called the principal component of variables. If the cumulative variance contribution rate of m principal components reaches a higher percentage (such as 85%), the data of m principal components will be retained. Eigenvalues > 1 is a necessary condition for a good model, and the selected principal component is meaningful. In our data analysis, there are three principal components with eigenvalues more than 1. The variance contribution rate was explained by 60.05%, 13.79% and 8.81%, respectively. Therefore, we plotted/used three.
Question 6: Table 2 header: lactone
Response: Thank you for your comment. Lactone: an organic compound containing an ester group —OCO— as part of a ring. It's a physicochemical parameter in the honey we measured.
Question 7: Fig.2. lower case letters showing significance seems to be mixed in subplot II. Why is IMH c, just like the sixth sample?
Response: Thank you for your comment. The mistake has been corrected in Table 2, and the Fig.2. II has been modified.
Question 8: L243: dispersal peace ? please revise.
Include the used statistical test in Table headers and Figure captions, where relevant.
Response: We appreciate your comment. The mistake has been corrected in line 248. And all Tables and Figures have been rechecked carefully.
Reviewer 2 Report
The article "Discrimination of natural mature acacia honey based on multi-physicochemical parameters combined with chemometric analysis"
deals with discrimination of mature/non mature acacia honeys using physicochemical parameters and chemometry.
Total phenolic content, total protein content and 16 total sugar (glucose, fructose, sucrose) were found as the major variables.
- The authors report total sugar (glucose, fructose, sucrose) as one of the major variables between mature and immature honey.
This could be explained by the water concentration. In fact, if one recalculate the values in table, e.g. the difference in sugar content results from
difference of water content (Table 2).
The following question remained without an answer:
What was the criteria and idea for grouping and comparison of the samples in Table 2? I think that 2
groups should be present: mature and immature samples, and significant differences between those 2 groups
should be investigated [considering te obvious difference of water content]. It is difficult to find out the significant differences in composition of
immature/immature honey that are not simply related to the moisture content, since the results in table seem not to be recalculated on dry weight.
Current methods for maturity evaluate sugar content (amount of sucrose - level of digestion), moisture and proline content
as marker of maturity - so these findings are not new.
- If the authors planned to distinguish between mature honey and dried immature honey to detect the unwanted honey manipulation mentioned in Apimondia document,
the samples from the same hive/combs should be collected at different time and the immature sample should be dried to similar moisture level before the comparison.
- Moreover, the authors claim to have corrected the description of the methodology. Meanwhile it is still incorrect:
AOAC Official Method 969.39 referred as test for glucose oxidase in fact is
Method used for determination of glucose in Corn Syrups and Dextrose
Products using Glucose Oxidase.
AOAC Official Method 983.15 referred as method for total phenolic content
is in fact a method used to determine phenolic antioxidants such as BHT, BHA etc.
in fats and oils by LC-UV.
actually I did not even find methods to determine such parameters (glucose oxidase, total phenolic content in honey among AOAC methods.
Author Response
Dear Reviewer, we are truly grateful to your critical comments and thoughtful suggestions.
The article "Discrimination of natural mature acacia honey based on multi-physicochemical parameters combined with chemometric analysis" deals with discrimination of mature/non mature acacia honeys using physicochemical parameters and chemometry.
Total phenolic content, total protein content and 16 total sugar (glucose, fructose, sucrose) were found as the major variables.
Question 1: The authors report total sugar (glucose, fructose, sucrose) as one of the major variables between mature and immature honey. This could be explained by the water concentration. In fact, if one recalculate the values in table, e.g. the difference in sugar content results from difference of water content (Table 2).
Response: Thank you for your comment. As per your suggestion, we rework the Table 2 on a dry weight basis. The data analysis you see now is to use the dry weight data to build the model. According to the results of PCA, total sugar has a greater impact on its classification (Table 3 and Figure 3). This is our conclusion by chemometrics.
Question 2: What was the criteria and idea for grouping and comparison of the samples in Table 2? I think that 2 groups should be present: mature and immature samples, and significant differences between those 2 groups should be investigated [considering te obvious difference of water content]. It is difficult to find out the significant differences in composition of immature/immature honey that are not simply related to the moisture content, since the results in table seem not to be recalculated on dry weight.
Response: Thank you for your suggestion. First, it is our fault that we did not clearly describe the Table 2, and this has now been described in detail in a new Table 2 in the manuscript. In new Table 2, A1-A29 represents mature honey and B30-B85 represents immature honey, the criteria and idea for grouping and comparison of the samples is whether honey is capped in the hive or not, and we have described them in detail in 2.1 honey samples.(Line 80-89).
Second, we rework Table 2 on a dry weight basis. The data analysis you see now is to use the dry weight data to build the model. We have modified the corresponding description as follows:
In order to avoid the influence of moisture on other physicochemical parameters, the moisture was eliminated when the model was established, and the data of other parameters were normalized, and used data on a dry-weight basis (Table 2).
Question 3:Current methods for maturity evaluate sugar content (amount of sucrose - level of digestion), moisture and proline content as marker of maturity - so these findings are not new.
- If the authors planned to distinguish between mature honey and dried immature honey to detect the unwanted honey manipulation mentioned in Apimondia document, the samples from the same hive/combs should be collected at different time and the immature sample should be dried to similar moisture level before the comparison.
Response: Thank you for your comments. First, our paper does not "shock the universe", and the introduction of the manuscript does not clearly explain the new ideas and innovations; in view of this, we have strengthened the introduction and emphasized the innovation (Line 62-65; Line 76-77). The innovation of this article is that we are the first to introduce China's mature honey and immature honey as well as use chemometric methods to distinguish them. In addition, the harvest of immature honey and its further processing in factories is not in accordance with Codex Alimentarius the accepted international standard for foods and a statement was made by APIMONDIA on honey fraud. Therefore, it is very meaningful to study China's mature honey and immature honey and identify mature honey.
Second, at present, our sample in this study are only mature honey and immature honey, but we have also analyzed the concentrated honey which is dried to similar moisture level, and identified them by chemometric analysis, and the results are satisfactory. We will sort out the relevant research as soon as possible. In addition, we hope you can give us valuable advice.
Question 4:Moreover, the authors claim to have corrected the description of the methodology. Meanwhile it is still incorrect:
AOAC Official Method 969.39 referred as test for glucose oxidase in fact is Method used for determination of glucose in Corn Syrups and Dextrose Products using Glucose Oxidase.
AOAC Official Method 983.15 referred as method for total phenolic content is in fact a method used to determine phenolic antioxidants such as BHT, BHA etc.in fats and oils by LC-UV. Actually I did not even find methods to determine such parameters (glucose oxidase, total phenolic content in honey among AOAC methods.
Response: Thank you for your comment. It is our fault that we did not seriously correct the mistake. The mistake has been corrected. Due to the length of the article and the requirements of the journal, we will simplify the method, the revised specific experimental method can be found in the supplementary materials, we hope you can understand.
Reviewer 3 Report
The study entitled “Discrimination of natural mature acacia honey based on multi-physicochemical parameters combined with chemometric analysis” needs a profound change in its structure and way of explaining the results. The approach is good, however this work must be thoroughly revised so that the readers can better understand the methodology and the results.
Here are some of the suggestions that could improve the understanding of this study.
Abstract:
- Line 14. Please changing “mature acacia honey and immature acacia honey” by mature and immature acacia honey.
- Line 15 to 17. This sentence is a little confuse “… were the major variables” for what?
Introduction:
- Line 37: Please, correct “the pr,oventriculus”.
- I think the justification for conducting this study is not clear. I suggest introducing a phrase that explains why mature honey is studied and why it is compared to non-mature honey. I suggest improving the next sentence” we analyzed 85 acacia honey samples which were collected from different regions of Shaanxi province in China, expecting that the method can be applied to quality control and authenticity identification of natural mature honey.”
Materials and methods (I suggest improving the material and methods section extensively)
- Line 81 to 82. The next sentence should be in the introduction section “Acacia honey is produced by honeybees which collect nectar from the flowers of Robinia pseudoacacia (Figure 1)”
- Line 83. What kind of classical quality determinations? Can the authors specify more?
- 2.2. Materials and reagents: I think this section is not necessary. You can explain the reagents used in the corresponded analysis or include this section in an annex.
- 2.3. Physicochemical properties:
I think this sentence is enough “according to the Association of Official Analytical Chemists (AOAC 1990).” the rest can be deleted.
Pollen analysis is not well detailed, the honey solution is not specified (with water?), the concentration of the honey? ... I suggest explaining it much better.
HPLC conditions or carbohydrates analysis by HPLC?
- I suggest improving the material and methods section extensively.
Results and discussion
- Please, follow the order of the sections of the previous section.
- I suggest explaining the results as a comparison between both types of honey (mature and not mature) and not repeating the values that are already exposed in the tables.
- In table 3 I suggest introducing a zero in front of the decimal point.
- In the PCA section, the conditions of the analysis are applied more than the result. Expand better which were the variables that most influence the separation of both types of honey.
Author Response
Dear Reviewers, we are truly grateful to your critical comments and thoughtful suggestions.
The study entitled “Discrimination of natural mature acacia honey based on multi-physicochemical parameters combined with chemometric analysis” needs a profound change in its structure and way of explaining the results. The approach is good, however this work must be thoroughly revised so that the readers can better understand the methodology and the results.
Here are some of the suggestions that could improve the understanding of this study.
Question 1:Line 14. Please changing “mature acacia honey and immature acacia honey” by mature and immature acacia honey.
Response: Thank you for your comment. We have modified in Line 14.
Question 2: Line 15 to 17. This sentence is a little confuse “… were the major variables” for what?
Response: We appreciate your comment. “Total phenolic content, total protein content and total sugar (glucose, fructose, sucrose) were the major variables”, this conclusion is obtained by PCA, according to Fig. 3B: loading, the loadings of each compound on the principal component analysis explicitly showed that the grouping of the different maturity honey was mainly influenced by certain compounds. The PC1 and PC2 of the all samples explained 73.83% of the total variance at length. PC1 was directly relevant to L*, total phenolic content, proline, and the dominant variables were mainly affected by protein, total phenol and sucrose in PC2. Moreover, combined with variance contribution rate of Table 3, we indicated that total phenolic content, total protein content and total sugar (glucose, fructose, sucrose) were the major variables.
Question 3:Line 37: Please, correct “the pr,oventriculus”.
Response: The mistake has been corrected in line 37.
Question 4: I think the justification for conducting this study is not clear. I suggest introducing a phrase that explains why mature honey is studied and why it is compared to non-mature honey. I suggest improving the next sentence” we analyzed 85 acacia honey samples which were collected from different regions of Shaanxi province in China, expecting that the method can be applied to quality control and authenticity identification of natural mature honey.”
Response: Thank you for your comment. We rewrote this part of the introduction (Lines 75-83).
Question 5: Materials and methods (I suggest improving the material and methods section extensively)
Line 81 to 82. The next sentence should be in the introduction section “Acacia honey is produced by honeybees which collect nectar from the flowers of Robinia pseudoacacia (Figure 1)”
Response: We appreciate your comment. This part was rewritten in Lines 29-32 as follows: Acacia honey is the natural sweet substance produced by honeybees, which collect nectar from the flowers of Robinia pesudoacacia (Figure 1), transform and combine it with specific substances of their own, store it, and leave it in the honeycomb to ripen and mature (Council Directive 2001/110/EC).
Question 6: Line 83. What kind of classical quality determinations? Can the authors specify more?
Response: Thank you for your comment. This information has been added in line 91-92.
Question 7: 2.2. Materials and reagents: I think this section is not necessary. You can explain the reagents used in the corresponded analysis or include this section in an annex.
Response: As per your suggestion, we have explained this section in the corresponded analysis.
Question 8:2.3. Physicochemical properties:
I think this sentence is enough “according to the Association of Official Analytical Chemists (AOAC 1990).” the rest can be deleted.
Pollen analysis is not well detailed, the honey solution is not specified (with water?), the concentration of the honey? ... I suggest explaining it much better.
HPLC conditions or carbohydrates analysis by HPLC? I suggest improving the material and methods section extensively.
Response: Thank you for your suggestion. First, we have modified the information of physicochemical properties in lines 117-120, as follows: The methods used for the quantitative analysis of physicochemical properties were determined mainly according to the Association of Official Analytical Chemists (AOAC 1990). The details of the methods used in this study are summarized in Supplementary material.
Second, pollen analysis has now been described in detail in Line 129-140 as follows: The botanical origin of the samples was determined by the method of Lutier and Vassiere (1993). For floral identification, honey samples (5 g) were thoroughly mixed with distilled water (5 mL), and centrifuged at 3000 rpm for 10 min, to separate the pollens. Samples of separated pollen grains were spread with the help of a brush on a slide containing a drop of lactophenol. The slides were examined microscopically at 45×magnification, using a bright-field microscope (Olympus,Tokyo). According to the different volumes, contours, grooves, holes and other characteristics of pollen morphology pictures of different varieties, pollen varieties were identified. A total of 40 horizons and a certain number of pollen grains were observed (the total number of pollen should be more than 100 grains).
Pollen content (%)= (a certain pollen number of 40 horizons/ a total pollen number of 40 horizons) x 100%
Third, the HPLC conditions have been added in lines 141-149 as follows: The contents of fructose, glucose and sucrose were determined by high-performance liquid chromatography (HPLC) and a refractive index detector (Shodex R1-201H, Shanghai, China). The column was a Waters Carbohydrate High Performance (4.6×250mm, 4μm; Waters). Honey samples (5 g) were thoroughly mixed with ultrapure water (60 mL), the total volume of the mixture was adjusted to 100 mL with acetonitrile. The mobile phase was 78% acetonitrile and 22% ultrapure water (v/v) using an isocratic method. The solutions were filtered through a 0.45μm membrane filter prior to use. The column was operated at 35℃ and detector pool temperature was 35℃, the flow-rate was 1.0 mL min–1, the injection volume was 15μL.
Question 9: Results and discussion
Please, follow the order of the sections of the previous section. I suggest explaining the results as a comparison between both types of honey (mature and not mature) and not repeating the values that are already exposed in the tables.
Response: Thank you for your suggestion. We have kept the order of the “Materials and methods” section of the “Results and discussion” section. What’s more, we modified the part of results and discussion in detail (seen in 3.2. Physicochemical Parameters Analysis).
Question 10: In table 3 I suggest introducing a zero in front of the decimal point.
Response: Thank you for your comment. We have modified the Table 3 as your suggestion.
Question 11: In the PCA section, the conditions of the analysis are applied more than the result. Expand better which were the variables that most influence the separation of both types of honey.
Response: Thank you for your comment. We modified the part of PCA in lines 222-244.
Reviewer 4 Report
the ms could be improved in particular for the quality of scientific presentation of the data.
I believe the conclusion section is poor and it is mandatory to modify it with more emphasis on obtained results. The introduction section is also poor and authors should modify it according to uptodate bibliography. English style should be checked.
Author Response
Question 1: the ms could be improved in particular for the quality of scientific presentation of the data.
I believe the conclusion section is poor and it is mandatory to modify it with more emphasis on obtained results. The introduction section is also poor and authors should modify it according to up to date bibliography. English style should be checked.
Response: We appreciate your comment. The sections of introduction and conclusion have been modified, and re-referenced some up to date bibliography. Full text was edited by the native English speaker, including English language, grammar, punctuation, spelling, and overall style.
We would like to extend our heartfelt thanks for your valuable suggestions to our paper. We have revised the paper in accordance with your suggestions. Again many thanks for your great efforts make to review our paper.
Yours sincerely,
Round 2
Reviewer 3 Report
The study entitled “Discrimination of natural mature acacia honey based on multi-physicochemical parameters combined with chemometric analysis” is rigorous enough to be published in the journal. However, I recommend that you check well the spelling mistakes throughout the text, like the example I put below.
- Line 69 "baesd on" be careful with this kind of errors
Author Response
Thank you for your comment. The mistake has been corrected in line 69. We have carefully checked the spelling of the full text, thank you very much for your suggestion.
This manuscript is a resubmission of an earlier submission. The following is a list of the peer review reports and author responses from that submission.
Round 1
Reviewer 1 Report
Dear authors,
The paper is much improved compared to the previous form. In its present form it is clearer and more understandable.
I suggest that it be published in its current form
Reviewer 2 Report
The article "Discrimination of natural mature acacia honey based on multi-physicochemical parameters combined with chemometric analysis"
deals with discrimination of mature/non mature acacia honeys using physicochemical parameters and chemometry.
Total phenolic content, total protein content and 16 total sugar (glucose, fructose, sucrose) were found as the major variables.
Current methods for maturity that evaluate sugar content (amount of sucrose - level of digestion), moisture and proline content as marker of maturity
(Codex Standard for Honey, 2001; Bogdanov et al. Int Honey Commission Honey quality and international regulatory
standards: review by the International Honey Commission. Bee World 1999, 80, 61-69.https://doi.org/10.1080/0005772X.1999.11099428).
Thiks should be also mentioned in the introduction. Considering this, it seems that there are not much novel findings in this paper, that were indicated.
What was the criteria and idea for grouping and comparison of the samples in Table 2? I think that 2 groups should be present: mature and immature samples,
and significant differences between those 2 groups should be investigated. It is difficult to find out the significant differences in composition of
mature/immasture honey that are not simply related to the moisture content, since the results in table seem not to be recalculated on dry weight.
Additionally the methodology description should be checked and corrected e.g. AOAC Official Method 920.181 referred as total phenolic content,
is in fact determination of ash ( http://www.eoma.aoac.org/methods/info.asp?ID=13136). AOAC Official Method 959.12 refers to glucose, not glucose oxidase.
Reviewer 3 Report
p { margin-bottom: 0.1in; line-height: 115%; background: transparent none repeat scroll 0% 0%; }Dear Authors,
I see significant improvements compared to the previous version of your manuscript, yet, I think your paper is still not suitable for publication in its current form. To help you improve the paper, my suggestions can be found below:
Major issues.
1., Moisture was correctly removed from the dataset before building the models. However, what should have also been done is correction of all measured values with dry weight content, to be able to express them on a dry weight basis. Despite using fresh weight is usually accepted in food sciences, as there is a strong negative correlation between moisture and the content values, statistical analysis results are still showing the effects of moisture change only.
Present the data in Table 2 on a dry weight basis (e.g. Free acid meq / kg dry matter). Use these data to redo the analyzes and revise the paper accordingly. Keep in mind that if the maturity can be accurately predicted by measuring water content alone, doing lots of tests and chemometrics does not make much sense.
2., A series of Mann–Whitney U tests should be done to determine whether there is a significant difference between immature and mature honeys with respect to the checked parameters. The p-values from this will support your major claim in L311-312.
3., Cluster heights do not directly mean anything (Fig. 4.). Cluster validity can be checked by - for example - bootstrap resampling, as in Ryota Suzuki and Hidetoshi Shimodaira: Hierarchical Clustering with P-Values via Multiscale Bootstrap Resampling. Alternatively, you can show the misclassification rate using a cluster-based model, as in 3.4. Given the OPLS-DA, the clustering is somewhat redundant, and can be removed from the paper.
4., Provide data on other floral mature/immature honeys' moisture content (and other parameters that turn out to be influential on classification) to be able to assess robustness of your classification approach. The novelty of your work needs to be more justified.
Best regards.